Taxonomic revision of the Malagasy members of the Nesomyrmex angulatus species group using the automated morphological species delineation protocol NC-PART-clustering

Csősz Sándor sandorcsosz2@gmail.com
Fisher Brian L.
Entomology, California Academy of Sciences , San Francisco, CA , USA
Mikheyev Alexander
Electronic publication date: 2016 Mar 10
Publication date: 2016
Volume: 4
Electronic Location ID: e1796
Received 2015 Dec 15; Accepted 2016 Feb 22
Copyright: ©2016 Csősz and Fisher
Copyright year: 2016
Copyright holder: Csősz and Fisher
License: This is an open access article distributed under the terms of the Creative Commons Attribution License, which permits unrestricted use, distribution, reproduction and adaptation in any medium and for any purpose provided that it is properly attributed. For attribution, the original author(s), title, publication source (PeerJ) and either DOI or URL of the article must be cited.
License URL: https://creativecommons.org/licenses/by/4.0/

Keywords: Morphometry, Gap statistic, Species delimitation, Exploratory analyses, Biogeography, Nesomyrmex, New species

Funding: National Science Foundation DEB-0072713 DEB-0344731 DEB-0842395 California Academy of Sciences Ernst Mayr Travel Grant This study was supported by the National Science Foundation under Grant No. DEB-0072713, DEB-0344731, and DEB-0842395. Finally, SC was supported by the Schlinger Fellowship at the California Academy of Sciences and an Ernst Mayr Travel Grants to the MCZ. The funders had no role in study design, data collection and analysis, decision to publish, or preparation of the manuscript.

==============================
Background. Applying quantitative morphological approaches in systematics research is a promising way to discover cryptic biological diversity. Information obtained through twenty-first century science poses new challenges to taxonomy by offering the possibility of increased objectivity in independent and automated hypothesis formation. In recent years a number of promising new algorithmic approaches have been developed to recognize morphological diversity among insects based on multivariate morphometric analyses. These algorithms objectively delimit components in the data by automatically assigning objects into clusters.

Method. In this paper, hypotheses on the diversity of the Malagasy Nesomyrmex angulatus group are formulated via a highly automated protocol involving a fusion of two algorithms, (1) Nest Centroid clustering (NC clustering) and (2) Partitioning Algorithm based on Recursive Thresholding (PART). Both algorithms assign samples into clusters, making the class assignment results of different algorithms readily inferable. The results were tested by confirmatory cross-validated Linear Discriminant Analysis (LOOCV-LDA).

Results. Here we reveal the diversity of a unique and largely unexplored fragment of the Malagasy ant fauna using NC-PART-clustering on continuous morphological data, an approach that brings increased objectivity to taxonomy. We describe eight morphologically distinct species, including seven new species: Nesomyrmex angulatus (Mayr, 1862), N. bidentatus sp. n., N. clypeatus sp. n., N. devius sp. n., N. exiguus sp. n., N. fragilis sp. n., N. gracilis sp. n., and N. hirtellus sp. n.. An identification key for their worker castes using morphometric data is provided.

Conclusions. Combining the dimensionality reduction feature of NC clustering with the assignment of samples into clusters by PART advances the automatization of morphometry-based alpha taxonomy.

Introduction

Madagascar, one of Earth’s biodiversity hotspots (Myers et al., 2000), has a unique and very diverse ant fauna with a high degree of micoendemism. Multifaceted efforts to discover ant diversity in the Malagasy region have collected sufficient data to support species-level taxonomy (Fisher, 2005).

On Madagascar, the high rate of diversity and possible cryptic species (i.e., genealogical lineages that cannot be convincingly separated using conventional morphological approaches, see Seifert, 2009) pose extraordinary challenges for biodiversity research. The authors estimate that many groups, such as the Malagasy Nesomyrmex, may contain ten times more species than was previously described. This dramatic increase in suspected species is due to a profusion of microendemic species. Our approach to this taxonomic challenges is to apply a quantitative morphological approach in combination with modern algorithms to delineate species by statistical means.

The Malagasy representatives of the genus previously had been classified into four lineages: angulatus-group, hafahafa-group, madecassus-group and sikorai-group. These species groups were defined by Csősz & Fisher (2015) based on salient morphological characteristics. The taxonomy of the hafahafa group has been clarified using quantitative morphology (Csősz & Fisher, 2015), proving the power of these methods to tackle cryptic diversity in tropical biomes.

Here the diversity of a unique and largely unexplored fragment of the Malagasy ant fauna, the Nesomyrmex angulatus group, is inferred via a highly automated protocol involving the fusion of two algorithms, Nest Centroid clustering (NC clustering) (Seifert, Ritz & Csősz, 2014) and Partitioning Algorithm based on Recursive Thresholding (PART) (Nilsen & Lingjaerde, 2013) using continuous morphometric data. NC clustering has proven efficient at pattern recognition within large and complex datasets (Csősz et al., 2014; Guillem, Drijfhout & Martin, 2014; Wachter et al., 2015) and PART makes assignments to objectively-defined clusters based on statistical thresholds (Nilsen et al., 2013; Tibshirani, Walther & Hastie, 2001).

Delimitations of clusters recognized by these exploratory analyses were tested via confirmatory Linear Discriminant Analysis (LDA) and Multivariate Ratio Extractor, MRA (Baur & Leuenberger, 2011) following the earlier protocol of Csősz & Fisher (2015).

Multivariate evaluation of morphological data has revealed that the N. angulatus species-group comprises eight well-outlined clusters in the Malagasy zoogeographical region, all representing species; of these, seven taxa are new to science. The eight species outlined, Nesomyrmex angulatus (Mayr, 1862), N. bidentatus sp. n., N. clypeatus sp. n., N. devius sp. n., N. exiguus sp. n., N. fragilis sp. n., N. gracilis sp. n., and N. hirtellus sp. n. are described or redefined here based on worker caste. We provide a combined key that includes both a traditional, character-based key and a numeric identification tool that helps readers resolve the most problematic cases.

The final species hypotheses are corroborated by qualitative morphological characters. Combining NC clustering and PART has proved to be an efficient method to automate species delimitation in insect taxonomy.

Material and Methods

Ant samples used in this study comply with the regulations for export and exchange of research samples outlined in the Convention on Biological Diversity and the Convention on International Trade in Endangered Species of Wild Fauna and Flora. For field work conducted in Madagascar, permits to research, collect, and export ants were obtained from the Ministry of Environment and Forest as part of an ongoing collaboration between the California Academy of Sciences and the Ministry of Environment and Forest, Madagascar National Parks and Parc Botanique et Zoologique de Tsimbazaza. Approval Numbers: No. 0142N/EA03/MG02, No. 340N-EV10/MG04, No. 69 du 07/04/06, No. 065N-EA05/MG11, No. 047N-EA05/MG11, No. 083N-A03/MG05, No. 206 MINENVEF/SG/DGEF/DPB/SCBLF, No. 0324N/EA12/MG03, No. 100 l/fEF/SG/DGEF/DADF/SCBF, No. 0379N/EA11/MG02, No. 200N/EA05/MG02. Authorization for export was provided by the Director of Natural Resources.

In the present study, 23 continuous morphometric traits were recorded in 378 worker individuals belonging to 266 nest samples collected in the Malagasy region.

The material is deposited in the following institutions, abbreviations after Evenhuis (2013): CASC (California Academy of Sciences, San Francisco, California, USA), MCZ (Museum of Comparative Zoology, Cambridge, Massachusetts, USA), NHMB (Naturhistorisches Museum, Basel, Switzerland), NHMW (Naturhistorisches Museum Wien, Austria), MHNG (Muséum d’Histoire Naturelle, Geneva, Switzerland).

The full list of material morphometrically examined in this revision is listed in Table S1 with unique specimen identifiers (e.g., CASENT0486461). Designation of type material with detailed label information is given in the type material investigated sections for each taxon.

All images and specimens used in this study are available online on AntWeb (http://www.antweb.org). Images are linked to their specimens via the unique specimen code affixed to each pin (CASENT0486461). Online specimen identifiers follow this format: http://www.antweb.org/specimen/CASENT0486461.

Digital color montage images were created using a JVC KY-F75 digital camera and Syncroscopy Auto-Montage software (version 5.0), or a Leica DFC 425 camera in combination with the Leica Application Suite software (version 3.8). Distribution maps were generated in R (R Core Team, 2015) via the ‘phylo.to.map’ function using package phytools (Revell, 2012).

Measurements were taken with a Leica MZ 12.5 stereomicroscope equipped with an ocular micrometer at a magnification of 100×. Measurements and indices are presented as arithmetic means with minimum and maximum values in parentheses. Body size dimensions are expressed in µm. Due to the abundance of worker specimens relative to queen and male specimens, the present revision is based on the worker caste only. Revision based on the study of the workers is further facilitated by the fact that the name-bearing type specimens of the vast majority of existing ant taxa belong to the worker caste. All measurements were made by the first author. For the definition of morphometric characters, earlier protocols (Csősz, Heinze & Mikó, 2015; Csősz & Fisher, 2015) were considered. Explanations and abbreviations for measured characters are as follows:

CL: Maximum cephalic length in median line. The head must be carefully tilted to the position providing the true maximum. Excavations of hind vertex and/or clypeus reduce CL (Fig. 1A).

Figure 1 Illustrations for morphometric characters of Nesomyrmex angulatus species group.

Head in dorsal view (A) with measurement lines for CL, CW, CWb, PoOC and SL; frontal region of the head dorsum (B) with measurement lines for FRS (red accessory lines and arrows identify the torular lamella) and Cdep; lateral view of mesosoma (C) with measurement line for ML; lateral view of propodeum, petiole, and postpetiole (D) with measurement lines for MPST, NOH, NOL, PPL, and SPST; dorsal view of mesosoma (E) with measurement lines for PSTI and MW; lateral view of propodeum, petiole, and postpetiole (F) with measurement lines for PEH, PEL, and PPH; dorsal view of propodeum, petiole, and postpetiole (G) with measurement lines for SPBA, SPTI, PEW, and PPW.

CW: Maximum width of the head. Includes compound eyes (Fig. 1A).

CWb: Maximum width of head capsule without the compound eyes. Measured just posterior of the eyes (Fig. 1A).

CS: Absolute cephalic size. The arithmetic mean of CL and CWb.

Cdep: Antero-median clypeal depression. Maximum depth of the median clypeal depression on its anterior contour line as it appears in fronto-dorsal view (Fig. 1B).

EL: Maximum diameter of the compound eye (not shown).

FRS: Frontal carina distance. Distance between the frontal carinae immediately caudal of the posterior intersection points between the frontal carinae and the torular lamellae. If these dorsal lamellae do not laterally surpass the frontal carinae, the deepest point of the scape corner pits may be taken as the reference line. These pits take up the inner corner of the scape base when the scape is directed caudally and produce a dark triangular shadow in the lateral frontal lobes immediately posterior to the dorsal lamellae of the scape joint capsule (Fig. 1B).

ML (Weber length): Mesosoma length from caudalmost point of propodeal lobe to transition point between anterior pronotal slope and anterior pronotal shield. Preferentially measured in lateral view; if the transition point is not well defined, use dorsal view and take the centre of the dark-shaded borderline between pronotal slope and pronotal shield as anterior reference point. In gynes: length from caudalmost point of propodeal lobe to the most distant point of steep anterior pronotal face (Fig. 1C).

MW: Mesosoma width. In workers MW is defined as the longest width of the pronotum in dorsal view excluding the pronotal spines (Fig. 1E).

MPST: Maximum distance from the center of the propodeal stigma to the anteroventral corner of the ventrolateral margin of the metapleuron (Fig. 1D).

NOH: Maximum height of the petiolar node. Measured in lateral view from the uppermost point of the petiolar node perpendicular to a reference line from the petiolar spiracle to the imaginary midpoint of the transition between dorso-caudal slope and dorsal profile of caudal cylinder of the petiole (Fig. 1D).

NOL: Length of the petiolar node. Measured in lateral view from the center of the petiolar spiracle to dorso-caudal corner of caudal cylinder. Do not take as the reference point the dorso-caudal corner of the helcium, which is sometimes visible (Fig. 1D).

PEH: Maximum petiole height. The chord of the ventral petiolar profile at node level is the reference line perpendicular to which the maximum height of petiole is measured (Fig. 1F).

PEL: Diagonal petiolar length in lateral view; measured from anterior corner of subpetiolar process to dorso-caudal corner of caudal cylinder (Fig. 1F).

PEW: Maximum width of petiole in dorsal view. Nodal spines are not considered (Fig. 1G).

PoOC: Postocular distance. Use a cross-scaled ocular micrometer and adjust the head to the measuring position of CL. Caudal measuring point: median occipital margin; frontal measuring point: median head at the level of the posterior eye margin (Fig. 1A).

PPH: Maximum height of the postpetiole in lateral view. Measured perpendicularly to a line defined by the linear section of the segment border between dorsal and ventral petiolar sclerite (Fig. 1F).

PPL: Postpetiole length. The longest anatomical line that is perpendicular to the posterior margin of the postpetiole and is between the posterior postpetiolar margin and the anterior postpetiolar margin (Fig. 1D).

PPW: Postpetiole width. Maximum width of postpetiole in dorsal view (Fig. 1G).

PSTI: Apical distance of pronotal spines in dorsal view; if spine tips are rounded or thick take the centers of spine tips as reference points (Fig. 1E).

SL: Scape length. Maximum straight line scape length excluding the articular condyle (Fig. 1A).

SPBA: Minimum spine distance. The smallest distance of the lateral margins of the spines at their base. This should be measured in dorsofrontal view, since the wider parts of the ventral propodeum do not interfere with the measurement in this position. If the lateral margins of spines diverge continuously from the tip to the base, a smallest distance at base is not defined. In this case, SPBA is measured at the level of the bottom of the interspinal meniscus (Fig. 1G).

SPST: Spine length. Distance between the center of propodeal stigma and spine tip. The stigma center refers to the midpoint defined by the outer cuticular ring but not to the center of the real stigma opening, which may be positioned eccentrically (Fig. 1E).

SPTI: Apical spine distance. The distance of spine tips in dorsal view; if spine tips are rounded or truncated, the centers of spine tips are taken as reference points (Fig. 1G).

Two characters (Cdep and PSTI) were used only in two species, N. angulatus and N. clypeatus sp. n., hence these characters were not involved in overall multivariate analyses.

Taxonomic nomenclature, OTU concepts, and natural language (NL) phenotypes were compiled in mx (http://purl.org/NET/mx-database). Taxonomic history and descriptions of taxonomic treatments were rendered from this software. Hymenoptera-specific terminology of morphological statements used in descriptions and identification key, and diagnoses are mapped to classes in phenotype-relevant ontologies (Hymenoptera Anatomy Ontology (HAO) (Yoder et al., 2010) via a URI table (Table S2); for more information about this approach see Seltmann et al. (2012) and Mikó et al. (2014).

In verbal descriptions of taxa based on external morphological traits, recent taxonomic papers (Csősz et al., 2014; Seifert & Csősz, 2015) were considered. Definitions of surface sculpturing are linked to Harris (1979). Body size is given in µm, means of morphometric ratios as well as minimum and maximum values are given in parentheses with up to three digits. Inclination of pilosity and cuticular spines is given in degrees. Definitions of species-groups as well as descriptions of species are surveyed in alphabetic order.

The electronic version of this article in Portable Document Format (PDF) will represent a published work according to the International Commission on Zoological Nomenclature (ICZN), and hence the new names contained in the electronic version are effectively published under that code from the electronic edition alone. This published work and the nomenclatural acts it contains have been registered in ZooBank, the online registration system for the ICZN. The ZooBank LSIDs (Life Science Identifiers) can be resolved and the associated information viewed through any standard web browser by appending the LSID to the prefix http://zoobank.org/. The LSID for this publication is: lsid:zoobank.org:pub:63B1A3E5-9E62-46AD-B594-6B3E83364D90. The online version of this work is archived and available from the following digital repositories: PeerJ, PubMed Central and CLOCKSS.

Statistical framework—hypothesis formation and testing

The present statistical framework follows the procedure applied in Csősz & Fisher (2015). Advantages and limitations of the present procedure are discussed there.

Data preparation and cleaning

Nest-centroid clustering (NC-clustering), and linear discriminant analysis (LDA) do not require special data preparation (e.g., standardization), hence raw data were applied for each of the statistical analyses. Data, however, are standardized (i.e., centered and scaled) for the multivariate ratio analysis (MRA) to prevent variables with large values from dominating the analysis (Baur & Leuenberger, 2011). Variables are tested via matrix scatterplots and Pearson product-moment correlation coefficients for error variance. The lack of a positive within-class correlation between different traits may indicate measurements errors, or may represent a morphological artifact (Baur et al., 2014). All traits but one, postpetiole length (PPL), have shown strong linear correlations to other traits. The distribution of PPL is more likely spherical, caused by an unknown source of error. For this reason, PPL was removed from further analyses. Raw data in µm is given in Table S3.

Generating prior species hypotheses via the combined application of NC clustering and PART

This method searches for discontinuities in continuous morphometric data and sorts all similar cases into the same cluster in a two-step procedure. The first step reduces dimensionality in data with cumulative linear discriminant analysis (LDA) using nest samples (i.e., individuals collected from the same nest are assumed genetically closely related, often sisters) as groups (Seifert, Ritz & Csősz, 2014). The second step calculates pairwise distances between samples using LD scores as input and the distance matrix is displayed in a dendrogram. The NC-clustering was done via packages cluster (Maechler et al., 2014) and MASS (Venables & Ripley, 2002).

The ideal number of clusters was determined by Partitioning Algorithm based on Recursive Thresholding via the package clusterGenomics (Nilsen & Lingjaerde, 2013) using the function ‘part’, which also assigns observations (i.e., specimens, or samples) into partitions. The method estimates the number of clusters in a data based on recursive application of the Gap statistic (Tibshirani, Walther & Hastie, 2001) and is able to discover both top-level clusters as well as sub-clusters nested within the main clusters. If more than one cluster is returned by the Gap statistic, it is re-optimized on each subset of cases corresponding to a cluster until a stopping threshold is reached or the subset under evaluation has less than 2*minSize cases (Nilsen et al., 2013). Two clustering methods are used to determine the optimal number of clusters “hclust” and “kmeans” with 1,000 bootstrap iterations. The results of PART are mapped on the dendrogram in colored bars via the function ‘mark.dendrogram’ found in Beleites & Sergo (2015). The script was written in R and can be found in Appendix S1.

Arriving at final species hypothesis using confirmatory Linear Discriminant Analysis (LDA) and LDA ratio extractor

To provide increased reliability of species delimitation, hypotheses for clusters and classification of cases via exploratory processes were confirmed by LDA Leave-one-out cross-validation (LOOCV). Classification hypotheses were imposed for all samples congruently classified by partitioning methods, while wild-card settings (i.e., no prior hypothesis imposed on the classification) were given to samples that were incongruently classified by the two methods or proved to be outliers. To extract the best ratios for the easiest species separation in the key and diagnoses we applied multivariate ratio analysis (MRA), a modern statistical method based on principal component analysis (PCA) and linear discriminant analysis (LDA) (Baur & Leuenberger, 2011).

Results and Discussion

Eight clusters were identified by both clustering algorithms ‘hclust’ and ‘kmeans’ using function ‘part’. The pattern recognized by these partitioning algorithms can be fitted on the hierarchical structure seen on the dendrogram generated by NC clustering (Fig. 2).

Figure 2 Dendrogram solution for Nesomyrmex angulatus species group.

Sample information in the dendrogram follows the given format: final species hypothesis confirmed by cross-validation LDA is followed by CASENT number separated by a hyphen. Final species hypothesis bar shows classification of samples after confirmation by cross-validated LDA. Different colors represent species. Nesomyrmex angulatus (Mayr, 1862): yellow, N. bidentatus sp. n.: red, N. clypeatus sp. n.: lilac, N. devius sp. n.: light blue, N. exiguus sp. n.: grey, N. fragilis sp. n.: green, N. gracilis sp. n.: dark blue, N. hirtellus sp. n.: brown. Prior species hypothesis was generated by method PART using two clustering methods, hclust (‘part-hclust’) and kmeans (‘part-kmeans’). Color code is the same as above, but outliers returned by ‘part-hclust’ appear in black.

The grouping hypotheses generated by the combination of hypothesis-free exploratory analyses was validated by Linear Discriminant Analysis with leave-one-out cross-validation (LOOCV-LDA). The overall classification success is 100% (Table 1). The phenetically distinguishable clusters represent eight morphologically diagnosable OTUs that differ in many qualitative characters (e.g., shape of propodeal spines, petiolar node, surface sculpturing, etc.), hence the eight clusters solution is accepted as the final species hypothesis.

Table 1 Classification matrix obtained by Leave One Out Cross Validation LDA.

The last column (percent.correct) shows the classification success in percentage.

	angulatus	bidentatus	clypeatus	devius	exiguus	fragilis	gracilis	hirtellus	percent. correct	
angulatus	33	0	0	0	0	0	0	0	100	
bidentatus	0	60	0	0	0	0	0	0	100	
clypeatus	0	0	12	0	0	0	0	0	100	
devius	0	0	0	27	0	0	0	0	100	
exiguus	0	0	0	0	84	0	0	0	100	
fragilis	0	0	0	0	0	42	0	0	100	
gracilis	0	0	0	0	0	0	44	0	100	
hirtellus	0	0	0	0	0	0	0	75	100	

The geographic distribution of each morphospecies corresponds to the known major areas of endemism in Madagascar (Brown et al., 2014; Vences et al., 2009) and can be characterized by one of the simplified bioclimatic zones of Madagascar (Schatz, 2000, after Cornet, 1974): eastern rainforest, central montane forest, western dry forest, and southwest desert spiny bush thicket (Fig. 3).

Figure 3 Dendrogram plotted on geographic map.

Allocation of species-pairs on maps and color codes for species are as follows: (A) Nesomyrmex bidentatus sp. n. (red) and N. fragilis sp. n. (blue); (B) N. devius sp. n. (green) and N. exiguus sp. n. (red); (C) N. hirtellus sp. n. (blue) and N. gracilis sp. n. (red); (D) N. angulatus (Mayr, 1862) (blue) and N. clypeautus sp. n. (red).

The eight species described here are as follows in alphabetic order: Nesomyrmex angulatus (Mayr, 1862), N. bidentatus sp. n., N. clypeatus sp. n., N. devius sp. n., N. exiguus sp. n., N. fragilis sp. n., N. gracilis sp. n., N. hirtellus sp. n..

These species are grouped into four species complexes based on morphological similarity. The bidentatus-complex consists of two species: Nesomyrmex bidentatus sp. n. and N. fragilis sp. n.; the devius-complex includes two new species: N. devius sp. n., N. exiguus sp. n., N. gracilis sp. n. and N. hirtellus sp. n.; while two species, N. angulatus (Mayr, 1862) and N. clypeatus sp. n., form a complex of their own in the Malagasy zoogeographical region. Separation of species as well as complexes are convincingly supported by Multivariate Ratio Analyses. Morphometric data for species calculated on individuals are given in Table 2.

Table 2 Mean of morphometric ratios calculated species-wise on individual level.

Morphometric traits are divided by absolute cephalic size (CS), ±SD are provided in the upper row, minimum and maximum values are given in parentheses in the lower row.

Species: nr. of individulals:	N. angulatus sp. n. (n = 33)	N. bidentatus sp. n. (n = 60)	N. clypeatus sp. n. (n = 12)	N. devius sp. n. (n = 27)	N. exiguus sp. n. (n = 84)	N. fragilis sp. n. (n = 42)	N. gracilis sp. n. (n = 44)	N. hirtellus sp. n. (n = 75)	
CS	691 ± 28.49	510 ± 27.4	898 ± 34	593 ± 18.7	586 ± 26.2	539 ± 31.4	620 ± 38.0	592 ± 26.5	
	[630, 727]	[419, 569]	[850, 946]	[562, 620]	[528, 644]	[469, 614]	[508, 699]	[525, 641]	
CL/CWb	1.259 ± 0.03	1.277 ± 0.03	1.076 ± 0.01	1.188 ± 0.02	1.213 ± 0.02	1.229 ± 0.02	1.199 ± 0.02	1.187 ± 0.02	
	[1.218, 1.327]	[1.204, 1.352]	[1.057, 1.105]	[1.147, 1.259]	[1.174, 1.255]	[1.189, 1.278]	[1.154, 1.246]	[1.142, 1.242]	
PoOC/CL	0.370 ± 0.01	0.416 ± 0.01	0.434 ± 0.01	0.394 ± 0.01	0.408 ± 0.01	0.404 ± 0.01	0.387 ± 0.01	0.387 ± 0.01	
	[0.355, 0.386]	[0.398, 0.440]	[0.423, 0.444]	[0.375, 0.408]	[0.391, 0.428]	[0.389, 0.421]	[0.372, 0.406]	[0.366, 0.404]	
FRS/CS	0.325 ± 0.01	0.399 ± 0.01	0.310 ± 0.01	0.415 ± 0.01	0.413 ± 0.01	0.409 ± 0.01	0.413 ± 0.01	0.412 ± 0.01	
	[0.310, 0.343]	[0.376, 0.419]	[0.300, 0.328]	[0.400, 0.428]	[0.397, 0.431]	[0.379, 0.430]	[0.390, 0.436]	[0.385, 0.427]	
SL/CS	0.815 ± 0.02	0.665 ± 0.02	0.758 ± 0.03	0.632 ± 0.01	0.656 ± 0.01	0.659 ± 0.02	0.647 ± 0.01	0.667 ± 0.02	
	[0.757, 0.866]	[0.634, 0.708]	[0.736, 0.835]	[0.616, 0.661]	[0.615, 0.683]	[0.615, 0.694]	[0.622, 0.685]	[0.611, 0.705]	
EL/CS	0.281 ± 0.01	0.264 ± 0.02	0.210 ± 0.01	0.263 ± 0.01	0.249 ± 0.01	0.260 ± 0.01	0.252 ± 0.01	0.272 ± 0.01	
	[0.262, 0.317]	[0.233, 0.311]	[0.193, 0.225]	[0.248, 0.279]	[0.228, 0.266]	[0.239, 0.276]	[0.228, 0.274]	[0.249, 0.289]	
MW/CS	0.673 ± 0.02	0.647 ± 0.01	0.699 ± 0.02	0.687 ± 0.01	0.684 ± 0.01	0.664 ± 0.01	0.693 ± 0.02	0.692 ± 0.01	
	[0.643, 0.699]	[0.621, 0.678]	[0.671, 0.732]	[0.658, 0.712]	[0.656, 0.728]	[0.627, 0.688]	[0.659, 0.726]	[0.664, 0.730]	
PEW/CS	0.407 ± 0.03	0.391 ± 0.02	0.460 ± 0.03	0.447 ± 0.01	0.437 ± 0.02	0.407 ± 0.02	0.454 ± 0.02	0.460 ± 0.02	
	[0.344, 0.451]	[0.330, 0.426]	[0.428, 0.512]	[0.422, 0.481]	[0.387, 0.480]	[0.363, 0.460]	[0.390, 0.487]	[0.409, 0.522]	
PPW/CS	0.486 ± 0.03	0.456 ± 0.01	0.493 ± 0.02	0.499 ± 0.02	0.516 ± 0.01	0.470 ± 0.02	0.500 ± 0.02	0.525 ± 0.02	
	[0.427, 0.546]	[0.425, 0.491]	[0.472, 0.521]	[0.464, 0.534]	[0.481, 0.548]	[0.429, 0.507]	[0.453, 0.539]	[0.475, 0.585]	
SPBA/CS	0.323 ± 0.02	0.346 ± 0.02	0.349 ± 0.02	0.371 ± 0.01	0.390 ± 0.02	0.369 ± 0.02	0.393 ± 0.02	0.389 ± 0.02	
	[0.265, 0.354]	[0.303, 0.372]	[0.326, 0.386]	[0.347, 0.402]	[0.352, 0.454]	[0.327, 0.405]	[0.350, 0.433]	[0.345, 0.427]	
SPTI/CS	0.332 ± 0.02	0.335 ± 0.02	0.463 ± 0.02	0.430 ± 0.01	0.436 ± 0.02	0.377 ± 0.02	0.489 ± 0.02	0.460 ± 0.02	
	[0.251, 0.375]	[0.303, 0.367]	[0.438, 0.489]	[0.401, 0.460]	[0.377, 0.493]	[0.337, 0.424]	[0.448, 0.536]	[0.418, 0.504]	
ML/CS	1.390 ± 0.03	1.338 ± 0.02	1.307 ± 0.03	1.256 ± 0.02	1.298 ± 0.02	1.308 ± 0.03	1.267 ± 0.02	1.315 ± 0.02	
	[1.302, 1.444]	[1.280, 1.379]	[1.257, 1.347]	[1.223, 1.285]	[1.214, 1.342]	[1.231, 1.345]	[1.201, 1.301]	[1.261, 1.379]	
PEL/CS	0.522 ± 0.02	0.571 ± 0.02	0.589 ± 0.02	0.543 ± 0.01	0.578 ± 0.02	0.567 ± 0.03	0.574 ± 0.02	0.585 ± 0.02	
	[0.482, 0.557]	[0.537, 0.605]	[0.558, 0.642]	[0.513, 0.565]	[0.496, 0.623]	[0.519, 0.717]	[0.541, 0.639]	[0.487, 0.645]	
NOL/CS	0.383 ± 0.02	0.321 ± 0.02	0.317 ± 0.02	0.304 ± 0.01	0.327 ± 0.01	0.317 ± 0.01	0.324 ± 0.01	0.324 ± 0.02	
	[0.317, 0.418]	[0.277, 0.362]	[0.290, 0.336]	[0.281, 0.332]	[0.279, 0.353]	[0.292, 0.347]	[0.278, 0.347]	[0.299, 0.407]	
PPL/CS	0.285 ± 0.01	0.280 ± 0.01	0.259 ± 0.01	0.298 ± 0.01	0.302 ± 0.01	0.286 ± 0.01	0.301 ± 0.01	0.315 ± 0.01	
	[0.250, 0.319]	[0.255, 0.305]	[0.231, 0.278]	[0.276, 0.313]	[0.274, 0.330]	[0.265, 0.308]	[0.280, 0.332]	[0.266, 0.333]	
SPST/CS	0.271 ± 0.02	0.264 ± 0.02	0.361 ± 0.01	0.340 ± 0.01	0.382 ± 0.02	0.310 ± 0.02	0.399 ± 0.03	0.375 ± 0.01	
	[0.223, 0.304]	[0.220, 0.335]	[0.335, 0.385]	[0.304, 0.356]	[0.317, 0.430]	[0.257, 0.356]	[0.301, 0.446]	[0.332, 0.416]	
MPST/CS	0.434 ± 0.01	0.428 ± 0.01	0.389 ± 0.01	0.395 ± 0.01	0.420 ± 0.01	0.420 ± 0.02	0.407 ± 0.01	0.425 ± 0.02	
	[0.385, 0.456]	[0.391, 0.458]	[0.353, 0.405]	[0.378, 0.416]	[0.394, 0.445]	[0.384, 0.449]	[0.374, 0.425]	[0.391, 0.463]	
PEH/CS	0.401 ± 0.02	0.387 ± 0.01	0.389 ± 0.02	0.419 ± 0.01	0.436 ± 0.01	0.406 ± 0.01	0.435 ± 0.01	0.436 ± 0.01	
	[0.362, 0.446]	[0.366, 0.424]	[0.348, 0.434]	[0.395, 0.439]	[0.406, 0.469]	[0.378, 0.431]	[0.417, 0.459]	[0.402, 0.479]	
NOH/CS	0.231 ± 0.01	0.219 ± 0.01	0.243 ± 0.02	0.244 ± 0.01	0.261 ± 0.01	0.235 ± 0.01	0.272 ± 0.01	0.273 ± 0.01	
	[0.203, 0.251]	[0.189, 0.249]	[0.226, 0.278]	[0.227, 0.268]	[0.236, 0.292]	[0.210, 0.257]	[0.250, 0.292]	[0.240, 0.310]	
PPH/CS	0.357 ± 0.01	0.355 ± 0.01	0.332 ± 0.02	0.370 ± 0.01	0.388 ± 0.01	0.364 ± 0.01	0.385 ± 0.01	0.400 ± 0.01	
	[0.333, 0.380]	[0.327, 0.381]	[0.275, 0.365]	[0.350, 0.384]	[0.356, 0.411]	[0.329, 0.394]	[0.354, 0.410]	[0.378, 0.438]	
PSTI/CS	0.689 ± 0.03	na.	0.773 ± 0.02	na.	na.	na.	na.	na.	
	[0.584, 0.736]		[0.733, 0.801]						
Cdep (μm)	na.	na.	19.245 ± 2.80	na.	na.	na.	na.	na.	
			[15.385, 23.077]						

Synopsis of Species of Nesomyrmex angulatus Group

angulatus (Mayr, 1862)

= angulatus ilgii (Forel, 1894)

= latinodis (Mayr, 1895)

= angulatus concolor (Santschi, 1914)

bidentatus Csősz & Fisher sp. n.

clypeatus Csősz & Fisher sp. n.

devius Csősz & Fisher sp. n.

exiguus Csősz & Fisher sp. n.

fragilis Csősz & Fisher sp. n.

gracilis Csősz & Fisher sp. n.

hirtellus Csősz & Fisher sp. n.

Key to workers of Malagasy Nesomyrmex angulatus group

Note: absolute size is given in µm, indexes are dimensionless values minimum and maximum values are given in brackets. Classification power between couplet based on a certain character is calculated and percent value is given in parentheses.

1. Median clypeal notch present (Fig. 4A): Cdep (µm) = 19 [15, 23]… clypeatus

- Median clypeal notch absent, anterior edge of clypeus intact and convex (Fig. 4B)...2

2. Anterolateral corner of pronotum angulate in dorsal wiew (Fig. 4C). Frontal carina narrow, scape longer: FRS/SL < 0.5 (100%)… angulatus

- Anterolateral corner of pronotum rounded in dorsal view (Fig. 4D). Frontal carina wide, scape shorter: FRS/SL > 0.5 (100%)… 3

3. Propodel spines short: SPST/CS = 0.286 [0.220, 0.356] (94.9%). In lateral view dorsal contour line of propodeal spine or tubercle continues in a flat transition into metasomal dorsum (Figs. 4E–4F). Postpetiole narrower, mesosoma longer: PPW/ML < 0.38 [0.317, 0.386] (96.8%). Combination of best ratios (PPW/ML and MPST/SPST) yields 99.7% of correct classification (see Fig. 5A)...4 (bidentatus complex)

- Propodeal spines longer and acute: SPST/CS = 0.378 [0.301, 0.446], (94.9%). In lateral view dorsal contour line of propodeal spine continues in bent transition to metasomal dorsum (Fig. 4G). Postpetiole wider, mesosoma shorter: PPW/ML > 0.36 [0.369, 0.440] (96.8%). Combination of best ratios (PPW/ML and CW/SPST) yields 99.4% of correct classification (see Fig. 5A)... 5 (devius complex)

4. Propodeal spine very short: SPST CS = 0.264 [0.220, 0.335] (84.2%), forming blunt tubercle (Fig. 4E). Postocular distance longer, apical spine distance shorter: PoOC/SPTI = 1.396 [1.233, 1.658] (93.1%). Combination of best ratios (PoOC/SPTI and ML/PEH) yields 97.1% of correct classification (see Fig. 5B)... bidentatus

- Propodeal spine moderately long: SPST CS = 0.307 [0.257, 0.356] (84.2%) and acute (Fig. 4F). Postocular distance longer, apical spine distance shorter: PoOC/SPTI = 1.185 [1.044, 1.341] (93.1%). Combination of best ratios (PoOC/SPTI and ML/PEH) yields 97.1% of correct classification (see Fig. 5B)... fragilis

5. North Madagascar only, north of −15° latitude… 6

- The middle and southern part of Madagascar, south of −15° latitude… 7

6. In profile, petiolar node rounded, leaning backward (Fig. 4H). Postocular distance longer, apical spine distance shorter: PoOC/SPTI = 1.029 [0.929, 1.179] (94.5%). Combination of best ratios (PoOC/SPTI and CWb/ML) yields 100% of classification success (see Fig. 5C)... exiguus

- In profile, petiolar node rectangular (Fig. 4I). Postocular distance shorter, apical spine distance longer: PoOC/SPTI = 0.866 [0.773, 0.965] (94.5%). Combination of best ratios (PoOC/SPTI and CWb/ML) yields 100% of classification success (see Fig. 5C)... gracilis

7. Postocular distance longer, apical spine distance shorter: PoOC/SPST = 1.261 [1.163, 1.382] (92.2%). Combination of best ratios (PoOC/SPST and MW/PPH) yields 100% of classification success (see Fig. 5D)... devius

- Postocular distance shorter, apical spine distance longer: PoOC/SPST = 1.122 [0.991, 1.250] (92.2%). Combination of best ratios (PoOC/SPST and MW/PPH) yields 100% of classification success (see Fig. 5D)... hirtellus

Description and redefinition of species

Nesomyrmex angulatus (Mayr, 1862:739) (Figs. 6A–6C, Table S1, Table 2.)

Figure 4 Diagnostic characters for workers.

Dorsal view of anterior clypeal notch, red accessory line shows the anterior contour line of the clypeus (A); dorsal view of an intact anterior clypeal border, red accessory line shows the anterior contour line of the clypeus (B); dorsal view of the anterior part of mesosoma, red accessory line shows the presence of antero-lateral angle of the mesosoma (C); dorsal view of the anterior part of mesosoma, red accessory line shows the absence of antero-lateral angle of the mesosoma (D); lateral view of propodeal denticle, red accessory line shows short propodeal spine (E); lateral view of acute propodeal spine, red accessory line shows flat transition of dorsal contour line of propodeal spine into metasomal dorsum (F); lateral view of acute propodeal spine, red accessory line shows bent transition of dorsal contour line of propodeal spine into metasomal dorsum (G); lateral view of petiole, red accessory line shows rounded, backward-leaning dorsal petiolar profile (H); lateral view of petiole, red accessory line shows rectangular dorsal petiolar profile (I).

Figure 5 First and second best morphometric ratios.

Scatterplots of the two most discriminating ratios between workers of Nesomyrmex bidentatus complex and N. devius complex (A); N. bidentatus sp. n. and N. fragilis sp. n. (B); N. exiguus sp. n. and N. gracilis sp. n. (C); N. devius sp. n. and N. hirtellus sp. n. (D).

Figure 6 Nesomyrmex angulatus non-type worker (CASENT0134948).

Head in full-face view (A), lateral view of the body (B), dorsal view of the body (C).

Type material investigated.

Leptothorax angulatus Mayr, 1862:739—“Sinai” [Egypt], collect. G.Mayr. Lectotype, designated by Bolton 1982: 324 (1w NHMW, CASENT0914922);

Leptothorax angulatus r. ilgii Forel, 1894:82—“r. L. ilgii Forel typus Harar (Ilg)” [Ethiopia] coll. Forel. Syntype (1w NHMG, CASENT0908991);

Leptothorax latinodis Mayr, 1895:130—“latinodis” G. Mayr Type, “Delagoa Bay Mozambiqe”, collect. G. Mayr. Holotype. (1w NHMW, CASENT0914925), [morphometrically not investigated due to fractured mesosoma];

Leptothorax angulatus var. concolor Santschi, 1914:107—“L. Goniothorax angulatus Mayr v. concolor Sant Type”, Cote d’Afrique or. angl. Ile de Mombasa Allaud & Jeannel Oct. 1911 St.3. Syntypes (2w NHMB, CASENT0912893);

Description of workers. Body color: yellow; brown. Body color pattern: concolorous; only clava darker. Absolute cephalic size (µm): 688 [630, 724], (n = 33). Cephalic length vs. maximum width of head capsule (CL/CWb): 1.258 [1.218, 1.327]. Postocular distance vs. cephalic length (PoOc/CL): 0.371 [0.359, 0.386]. Postocular sides of cranium contour frontal view orientation: converging posteriorly. Postocular sides of cranium contour frontal view shape: feebly convex. Vertex contour line in frontal view shape: straight; feebly convex. Vertex sculpture: main sculpture rugoso-reticulate, ground sculpture areolate. Gena contour line in frontal view shape: convex. Genae contour from anterior view orientation: converging; strongly converging. Gena sculpture: rugoso-reticulate with areolate ground sculpture. Concentric carinae laterally surrounding antennal foramen count: present. Eye length vs. absolute cephalic size (EL/CS): 0.280 [0.262, 0.317]. Frontal carina distance vs. absolute cephalic size (FRS/CS): 0.325 [0.310, 0.343]. Longitudinal carinae on median region of frons: present. Smooth median region on frons: absent. Antennomere count: 12. Scape length vs. absolute cephalic size (SL/CS): 0.818 [0.783, 0.866]. Facial area of the scape absolute setal angle: setae absent, pubescence only. Median clypeal notch: absent. Ground sculpture of submedian area of clypeus: smooth. Median carina of clypeus: present. Lateral carinae of clypeus: present. Median anatomical line of propodeal spine angle value to Weber length in lateral view: 58–62°. Spine length vs. absolute cephalic size (SPST/CS): 0.273 [0.225, 0.304]. Minimum spine distance vs. absolute cephalic size (SPBA/CS): 0.325 [0.299, 0.354]. Apical spine distance vs. absolute cephalic size (SPTI/CS): 0.334 [0.294, 0.360]. Propodeal spine shape: straight; curving upward. Anterolateral pronotal corner: present. Apical distance of pronotal spines vs. absolute cephalic size (PSTI/CS): 0.690 [0.584, 0.736]. Metanotal depression: absent. Dorsal region of mesosoma sculpture: rugulose with areolate ground sculpture. Lateral region of pronotum sculpture: areolate ground sculpture, superimposed by dispersed rugae. Mesopleuron sculpture: areolate ground sculpture superimposed by dispersed rugulae. Metapleuron sculpture: areolate ground sculpture superimposed by dispersed rugulae. Petiole width vs. absolute cephalic size (PEW/CS): 0.410 [0.376, 0.445]. Dorsal region of petiole sculpture: ground sculpture areolate, main sculpture rogoso-reticulate. Postpetiole width vs. absolute cephalic size (PPW/CS): 0.488 [0.443, 0.546]. Dorsal region of postpetiole sculpture: ground sculpture areolate, main sculpture dispersed rugose.

Diagnosis. Workers of N. angulatus can be convincingly separated from those of N. clypeatus based on the lack of median clypeal notch in the former species (Fig. 4A). Nesomyrmex angulatus differs from species of bidentatus-complex and devius-complex (N. devius, N. exiguus, N. fragilis, N. gracilis and N. hirtellus) by having sharp anterolateral pronotal angles (Fig. 4C) and a numeric key, FRS/CS ratio yields perfect separation between workers of N. angulatus and members of bidentatus-complex (Table 2).

Distribution. In the Malagasy zoogeographical region, this species is known to occur in coastal dry forests, mangroves and the coastal scrub of the northern, dry area of Madagascar and on adjacent islands in the Mozambique channel (Fig. 3). Worldwide, N. angulatus has spread to the eastern Africa and the Middle East.

Nesomyrmex bidentatus Csősz & Fisher sp. n.

(Figs. 7A–7C, Table S1, Table 2.)

Figure 7 Nesomyrmex bidentatus sp. n. holotype worker (CASENT0486461).

Head in full-face view (A), lateral view of the body (B), dorsal view of the body (C).

Type material investigated.

Holotype worker: MADAGASCAR: Prov. Mahajanga, P N Namoroka, 16.9 km 317° NW Vilanandro, 16°24.4′S, 45°18.6′E, 100 m, 12-16.xi.2002, collection code: BLF6646; CASENT0486461, Fisher et al. (CASENT0486461, CAS);

Paratypes: Eighteen workers, a single gyne and a male with the same locality data under CASENT codes: CASENT0486459, BLF6646, (3w, CAS); CASENT0486460, BLF6646, (1m, 1q, CAS); CASENT0486462, BLF6646, (3w, CAS); CASENT0486797, BLF6618, (3w, CAS); CASENT0486798, BLF6618, (3w, CAS); CASENT0486799, BLF6618, (3w, CAS); CASENT0488445, BLF6584(24), (1w, CAS); CASENT0746773, BLF6646, (2w, CAS);

Description of workers. Body color: yellow. Body color pattern: concolorous. Absolute cephalic size (µm): 510 [419, 569] (n = 60). Cephalic length vs. maximum width of head capsule (CL/CWb): 1.277 [1.204, 1.352]. Postocular distance vs. cephalic length (PoOc/CL): 0.416 [0.398, 0.440]. Postocular sides of cranium contour frontal view orientation: converging anteriorly; parallel. Postocular sides of cranium contour frontal view shape: straight; feebly convex; convex. Vertex contour line in frontal view shape: straight; feebly convex. Vertex sculpture: main sculpture rugoso-reticulate, ground sculpture areolate. Gena contour line in frontal view shape: convex. Genae contour from anterior view orientation: converging; strongly converging. Gena sculpture: rugoso-reticulate with areolate ground sculpture. Concentric carinae laterally surrounding antennal foramen: absent. Eye length vs. absolute cephalic size (EL/CS): 0.264 [0.233, 0.311]. Frontal carina distance vs. absolute cephalic size (FRS/CS): 0.399 [0.376, 0.419]. Longitudinal carinae on median region of frons: absent. Smooth median region on frons: absent. Antennomere count: 12. Scape length vs. absolute cephalic size (SL/CS): 0.665 [0.634, 0.708]. Facial area of the scape absolute setal angle: 0–15°. Median clypeal notch: absent. Ground sculpture of submedian area of clypeus: present. Median carina of clypeus: present. Lateral carinae of clypeus: present. Median anatomical line of propodeal spine angle value to Weber length in lateral view: cannot be measured. Spine length vs. absolute cephalic size (SPST/CS): 0.264 [0.220, 0.335]. Minimum spine distance vs. absolute cephalic size (SPBA/CS): 0.346 [0.220, 0.335]. Apical spine distance vs. absolute cephalic size (SPTI/CS): 0.335 [0.303, 0.367]. Propodeal spine shape: triangular, blunt. Anterolateral pronotal corner: absent. Metanotal depression count: absent; inconspicuous if present. Dorsal region of mesosoma sculpture: rugulose with areolate ground sculpture. Lateral region of pronotum sculpture: areolate ground sculpture, superimposed by dispersed rugae. Mesopleuron sculpture: areolate ground sculpture, superimposed by dispersed rugae. Metapleuron sculpture: areolate ground sculpture, superimposed by dispersed rugae. Petiole width vs. absolute cephalic size (PEW/CS): 0.397 [0.331, 0.442]. Dorsal region of petiole sculpture: ground sculpture areolate, main sculpture dispersed rugose. Postpetiole width vs. absolute cephalic size (PPW/CS): 0.462 [0.425, 0.508]. Dorsal region of postpetiole sculpture: ground sculpture areolate, main sculpture dispersed rugose.

Etymology. The name (bidentatus) refers to the short propodeal denticle pair of this species.

Diagnosis. Workers of N. bidentatus differ from those of N. clypeatus by having no median clypeal notch (Fig. 4B) and from N. angulatus by the lack of an anterolateral pronotal corner (Fig. 4D). This species can be well separated from N. devius, N. exiguus, N. gracilis, and N. hirtellus based on its short and blunt spines and shorter apical spine distance (SPST/CS, see Table 2). This species is the most similar to N. fragilis: these two species can be separated by PooC/SPTI ratio, which yields 92.9% classification success (Fig. 5B). Due to the fact that the MRA plot offers only 97.3% of the correct classification, a reduced discriminant function using a combination of four characters (D4 = 0.062581 ML − 0.052596 CW − 0.095374 SPBA − 0.042818 SPST + 6.642672) that yields >99% classification success is also given to provide the most accurate determination in areas where the two species occur syntopically.

D4 scores for single individuals:

T. bidentatus sp. n. (n = 60) D4 = + 1.973 [−0.984, 4.366]

T. fragilis sp. n. (n = 42) D4 = − 2.234 [−4.417, −0.592]

Distribution. Nesomyrmex bidentatus is distributed in rainforests and littoral rainforests along the coastline around the entirety of Madagascar. This species occurs syntopically with its sister species N. fragilis in the western Antsisarana region (Fig. 3).

Nesomyrmex clypeatus Csősz & Fisher sp. n.

(Figs. 8A–8C, Table S1, Table 2.)

Figure 8 Nesomyrmex clypeatus sp. n. holotype worker (CASENT0422552).

Head in full-face view (A), lateral view of the body (B), dorsal view of the body (C).

Type material investigated.

Holotype worker: MADGAGASCAR: Prov. Antsiranana, Rés. Spéc. Ankarana, 22.9 km 224° SW Anivorano Nord, 12°55′S, 49°07′E, 80 m, 10-16. ii.2001, collection code: BLF3004; CASENT0422552, Fisher et al. (CASENT0422552, CAS);

Paratypes: four workers with the same locality data under CASENT codes: CASENT0427944, BLF3007, (1w, CAS); CASENT0422553, BLF2968, (1w, CAS); CASENT0427970, BLF2970, (1w, CAS); CASENT0427971, BLF2971, (1w, CAS);

Description of workers. Body color: yellow; brown. Body color pattern: concolorous, only clava darker. Absolute cephalic size (µm): 898 [850, 946] (n = 12). Cephalic length vs. maximum width of head capsule (CL/CWb): 1.076 [1.057, 1.105]. Postocular distance vs. cephalic length (PoOc/CL): 0.434 [0.423, 0.444]. Postocular sides of cranium contour frontal view orientation: converging posteriorly. Postocular sides of cranium contour frontal view shape: feebly convex. Vertex contour line in frontal view shape: straight; feebly convex. Vertex sculpture: main sculpture rugoso-reticulate, ground sculpture areolate. Gena contour line in frontal view shape: convex. Genae contour from anterior view orientation: strongly converging. Gena sculpture: rugoso-reticulate with areolate ground sculpture. Concentric carinae laterally surrounding antennal foramen: absent. Eye length vs. absolute cephalic size (EL/CS): 0.210 [0.193, 0.225]. Frontal carina distance vs. absolute cephalic size (FRS/CS): 0.310 [0.300, 0.328]. Longitudinal carinae on median region of frons: absent. Smooth median region on frons count: absent. Antennomere count: 12. Scape length vs. absolute cephalic size (SL/CS): 0.758 [0.736, 0.835]. Facial area of the scape absolute setal angle: setae absent, pubescence only. Median clypeal notch: present. Median clypeal notch depth vs. absolute cephalic size (Cdep/CS): 0.021 [0.013, 0.027]. Ground sculpture of submedian area of clypeus: present. Median carina of clypeus: present. Lateral carinae of clypeus: present. Median anatomical line of propodeal spine angle value to Weber length in lateral view: 50–60°. Spine length vs. absolute cephalic size (SPST/CS): 0.361 [0.335, 0.385]. Minimum spine distance vs. absolute cephalic size (SPBA/CS): 0.349 [0.326, 0.386]. Apical spine distance vs. absolute cephalic size (SPTI/CS): 0.463 [0.438, 0.489]. Propodeal spine shape: straight; slightly bent. Anterolateral pronotal corner: present. Apical distance of pronotal spines vs. absolute cephalic size (PSTI/CS): 0.773 [0.733, 0.801]. Metanotal depression count: absent. Dorsal region of mesosoma sculpture: rugose with areolate ground sculpture. Lateral region of pronotum sculpture: areolate ground sculpture, superimposed by dispersed rugae. Mesopleuron sculpture: areolate ground sculpture superimposed by dispersed rugulae. Metapleuron sculpture: areolate ground sculpture superimposed by dispersed rugulae. Petiole width vs. absolute cephalic size (PEW/CS): 0.460 [0.428, 0.512]. Dorsal region of petiole sculpture: ground sculpture areolate, main sculpture rogoso-reticulate. Postpetiole width vs. absolute cephalic size (PPW/CS): 0.493 [0.472, 0.521]. Dorsal region of postpetiole sculpture: ground sculpture areolate, main sculpture dispersed rugose.

Etymology. The name (clypeatus) refers to the presence of an antero-median clypeal depression in this species, the characteristic found to be unique in this revisionary work.

Diagnosis. This species cannot be confused with other taxa in this revisionary work based on the dark antennal club and the conspicuous median notch (Cdep/CS: 0.021 [0.013, 0.027]) on the anterior clypeal border.

Distribution. This species is endemic to the Malagasy region. It is known to occur in tropical dry forests and littoral forests of the northern, dry area of Madagascar (Fig. 3).

Nesomyrmex devius Csősz & Fisher sp. n.

(Figs. 9A–9C, Table S1, Table 2.)

Figure 9 Nesomyrmex devius sp. n. holotype worker (CASENT0448820).

Head in full-face view (A), lateral view of the body (B), dorsal view of the body (C).

Type material investigated.

Holotype worker: MADGAGASCAR: Prov. Toliara, Mahafaly Plateau, Isantoria Riv., 6.2 km 74° ENE Itampolo, 24°39′S, 43°69′E, 80 m, 21-25.ii.2002, collection code: BLF5777; CASENT0448820, Fisher et al. (CASENT0448820, CAS);

Paratypes: fifteen workers, and 6 gynes with the same label data with the holotype under CASENT codes: CASENT0448818, BLF5777, (1w, CAS); CASENT0448819, BLF5777, (1w, CAS); CASENT0448823, BLF5777, (3w, CAS); CASENT0448824, BLF5777, (3w, CAS); CASENT0448825, BLF5777, (3w, CAS); CASENT0448829, BLF5777, (1q, CAS); CASENT0448830, BLF5777, (1q, CAS); CASENT0448831, BLF5777, (1q, CAS); CASENT0448832, BLF5777, (3q, CAS); CASENT0448833, BLF5777, (2w, CAS); CASENT0746772, BLF5777, (2w, CAS);

Description of workers. Body color: yellow; brown. Body color pattern: concolorous. Absolute cephalic size (µm): 593 [562, 620] (n = 27). Cephalic length vs. maximum width of head capsule (CL/CWb): 1.188 [1.147, 1.259]. Postocular distance vs. cephalic length (PoOc/CL): 0.394 [0.375, 0.408]. Postocular sides of cranium contour frontal view orientation: converging anteriorly; parallel. Postocular sides of cranium contour frontal view shape: straight; feebly convex; convex. Vertex contour line in frontal view shape: straight; feebly convex. Vertex sculpture: main sculpture rugose, ground sculpture areolate. Gena contour line in frontal view shape: convex. Genae contour from anterior view orientation: converging; strongly converging. Gena sculpture: rugoso-reticulate with areolate ground sculpture. Concentric carinae laterally surrounding antennal foramen: absent. Eye length vs. absolute cephalic size (EL/CS): 0.263 [0.248, 0.279]. Frontal carina distance vs. absolute cephalic size (FRS/CS): 0.415 [0.400, 0.428]. Longitudinal carinae on median region of frons: absent. Smooth median region on frons: absent. Antennomere count: 12. Scape length vs. absolute cephalic size (SL/CS): 0.632 [0.616, 0.661]. Facial area of the scape absolute setal angle: 0–15°. Median clypeal notch: absent. Ground sculpture of submedian area of clypeus: present. Median carina of clypeus: present. Lateral carinae of clypeus: present. Median anatomical line of propodeal spine angle value to Weber length in lateral view: 42–47°. Spine length vs. absolute cephalic size (SPST/CS): 0.340 [0.304, 0.356]. Minimum spine distance vs. absolute cephalic size (SPBA/CS): 0.371 [0.347, 0.402]. Apical spine distance vs. absolute cephalic size (SPTI/CS): 0.430 [0.401, 0.460]. Propodeal spine shape: straight. Anterolateral pronotal corner: absent. Metanotal depression count: present. Dorsal region of mesosoma sculpture: rugulose with areolate ground sculpture. Lateral region of pronotum sculpture: areolate ground sculpture, superimposed by dispersed rugae. Mesopleuron sculpture: areolate ground sculpture superimposed by dispersed rugulae. Metapleuron sculpture: areolate ground sculpture superimposed by dispersed rugulae. Petiole width vs. absolute cephalic size (PEW/CS): 0.447 [0.422, 0.481]. Dorsal region of petiole sculpture: ground sculpture areolate, main sculpture dispersed rugose; ground sculpture areolate, main sculpture rugoso-reticulate. Postpetiole width vs. absolute cephalic size (PPW/CS): 0.499 [0.464, 0.534]. Dorsal region of postpetiole sculpture: ground sculpture areolate, main sculpture dispersed rugose.

Etymology. The name (dēvius = devious) refers to the relatively long path required to arrive at the current taxonomic situation of this species, caused by its superficial similarities to other taxa.

Diagnosis. Workers of N. devius differ from those of N. clypeatus by having no median clypeal notch (Fig. 4B) and from those of N. angulatus by the lack of an anterolateral pronotal corner (Fig. 4D). This species can be separated from N. bidentatus and N. fragilis based on the apical spine distance ratio (SPTI/CS, see Table 2). This species occurs in the southern part of Madagascar syntopically with N. hirtellus from the N. devius complex. A simple ratio (PoOC/ SPST, see details in key) offers 92.2% success in determination between this species and N. Hirtellus, but a combination of two ratios (PoOC/SPST and MW/PPH) yields a safer determination (Fig. 5D). The other two species of this complex, N. exiguus and N. gracilis do not occur syntopically with this species, as these are distributed far to the north of the distributional area of N. devius.

Distribution. This species is endemic to the Malagasy region, and its distribution is restricted in the southwestern, sub-arid area of Madagascar (Fig. 3) occurring in rupicolous vegetation on granite outcrops and spiny forests.

Nesomyrmex exiguus Csősz & Fisher sp. n.

(Figs. 10A–10C, Table S1, Table 2.)

Figure 10 Nesomyrmex exiguus sp. n. holotype worker (CASENT0077581).

Head in full-face view (A), lateral view of the body (B), dorsal view of the body (C).

Type material investigated.

Holotype worker: MADAGASCAR: Prov. Antsiranana, Fort Antsahabe, 11.4 km 275° W Daraina, 13°13.7′S, 49°33.4′E, 550 m, 12-14.xii.2003, collection code: BLF10161; CASENT0077581, Fisher et al. (CASENT0077581, CAS);

Paratypes: seventeen workers, a single gyne and a male with the same locality data under CASENT codes: CASENT0077580, BLF10161, (1m, 1w, CAS); CASENT0746774, BLF10161, (1q, 1w, CAS); CASENT0077624, BLF10190, (3w, CAS); CASENT0077625, BLF10190, (3w, CAS); CASENT0077626, BLF10190, (3w, CAS); CASENT0077586, BLF10206, (3w, CAS); CASENT0077587, BLF10206, (3w, CAS);

Description of workers. Body color: yellow; brown. Body color pattern: concolorous. Absolute cephalic size (µm): 586 [528, 644] (n = 84). Cephalic length vs. maximum width of head capsule (CL/CWb): 1.213 [1.174, 1.255]. Postocular distance vs. cephalic length (PoOc/CL): 0.408 [0.391, 0.428]. Postocular sides of cranium contour frontal view orientation: parallel; converging anteriorly. Postocular sides of cranium contour frontal view shape: straight; feebly convex; convex. Vertex contour line in frontal view shape: straight; feebly convex. Vertex sculpture: main sculpture rugose, ground sculpture areolate. Gena contour line in frontal view shape: convex. Genae contour from anterior view orientation: converging; strongly converging. Gena sculpture: rugoso-reticulate with areolate ground sculpture. Concentric carinae laterally surrounding antennal foramen: absent. Eye length vs. absolute cephalic size (EL/CS): 0.249 [0.228, 0.266]. Frontal carina distance vs. absolute cephalic size (FRS/CS): 0.413 [0.397, 0.431]. Longitudinal carinae on median region of frons: absent. Smooth median region on frons: absent. Antennomere count: 12. Scape length vs. absolute cephalic size (SL/CS): 0.656 [0.615, 0.683]. Facial area of the scape absolute setal angle: 0–15°. Median clypeal notch: absent. Ground sculpture of submedian area of clypeus: present. Median carina of clypeus: present. Lateral carinae of clypeus: present. Median anatomical line of propodeal spine angle value to Weber length in lateral view: 27–32°. Spine length vs. absolute cephalic size (SPST/CS): 0.382 [0.317, 0.430]. Minimum spine distance vs. absolute cephalic size (SPBA/CS): 0.390 [0.352, 0.454]. Apical spine distance vs. absolute cephalic size (SPTI/CS): 0.436 [0.377, 0.493]. Propodeal spine shape: straight; slightly bent. Anterolateral pronotal corner: absent. Metanotal depression count: present. Dorsal region of mesosoma sculpture: rugulose with areolate ground sculpture. Lateral region of pronotum sculpture: areolate ground sculpture, superimposed by dispersed rugae. Mesopleuron sculpture: areolate ground sculpture superimposed by dispersed rugulae. Metapleuron sculpture: areolate ground sculpture superimposed by dispersed rugulae. Petiole width vs. absolute cephalic size (PEW/CS): 0.437 [0.387, 0.480]. Dorsal region of petiole sculpture: ground sculpture areolate, main sculpture dispersed rugose. Postpetiole width vs. absolute cephalic size (PPW/CS): 0.516 [0.481, 0.548]. Dorsal region of postpetiole sculpture: ground sculpture areolate, main sculpture dispersed rugose.

Etymology. This name exiguus (=strict, exact) refers to the fact that this species is relatively easily to distinguish.

Diagnosis. Workers of N. exiguus differ from those of N. clypeatus by having no median clypeal notch (Fig. 4B) and from those of N. angulatus by the lack of anterolateral pronotal corner (Fig. 4D). No single ratio separates this species from N. bidentatus and N. fragilis, but a combined application of two morphometric ratios (PPW/ML and CW/SPST) provides a safe opportunity for separation (Fig. 5A). This species occurs in the northern part of Madagascar syntopically wih N. gracilis from the N. devius complex. A simple ratio (PoOC/SPTI, see details in key) offers 94.5% success discriminating between this species and N. gracilis, and a combination of two ratios (PoOC/SPTI and CWb/ML) yields a safe determination (Fig. 5C). The other two species of this complex, N. devius and N. hirtellus, do not occur syntopically with this species, as these are distributed far south of the distributional area of N. exiguus.

Distribution. This species is endemic to the Malagasy region, and its distribution is restricted to the northern, dry area of Madagascar (Fig. 3). There it lives in littoral rainforest and tropical dry forest; a single locality (Forêt d’ Andavakoera) is known in rainforest close to other known localities of this species in the northern, dry bioclimatic zone. Two samples that may raise the chance of misclassifications are known to have been collected in far southern localities (Fig. 3). These samples were classified as N. exiguus by cumulative LDA with very high posterior probabilities (CASENT0208857, p = 0.971 and CASENT0496931, p = 0.998) when these were added as wildcards to minimize the chance of possible misclassifications. These individuals are most probably representatives of populations brought to these localities by people.

Nesomyrmex fragilis Csősz & Fisher sp. n.

(Figs. 11A–11C, Table S1, Table 2.)

Figure 11 Nesomyrmex fragilis sp. n. holotype worker (CASENT0421396).

Head in full-face view (A), lateral view of the body (B), dorsal view of the body (C).

Type material investigated.

Holotype worker: MADGAGASCAR: Prov. Antsisarana, Nosy Be, Réserve Naturelle Intégrale de Lokobe, 6.3 km 112° ESE Hellville, 13.41933°S, 48.33117°E, 30 m, 19-24.iii.2001, collection code: BLF3496; CASENT0421396, Fisher et al. (CASENT0421396, CAS);

Paratypes: five workers and a single gyne with the same locality data under CASENT codes: CASENT0421397, BLF3496, (1q, CAS); CASENT0421395, BLF3496, (1w, CAS); CASENT0421398, BLF3482, (2w, CAS); CASENT0421399, BLF3482, (2w, CAS);

Description of workers. Body color: yellow. Body color pattern: concolorous. Absolute cephalic size (µm): 539 [469, 614] (n = 42). Cephalic length vs. maximum width of head capsule (CL/CWb): 1.229 [1.189, 1.278]. Postocular distance vs. cephalic length (PoOc/CL): 0.404 [0.389, 0.421]. Postocular sides of cranium contour frontal view orientation: converging anteriorly; parallel. Postocular sides of cranium contour frontal view shape: straight; feebly convex; convex. Vertex contour line in frontal view shape: straight; feebly convex. Vertex sculpture: main sculpture rugoso-reticulate, ground sculpture areolate. Gena contour line in frontal view shape: convex. Genae contour from anterior view orientation: converging; strongly converging. Gena sculpture: rugoso-reticulate with areolate ground sculpture. Concentric carinae laterally surrounding antennal foramen: absent. Eye length vs. absolute cephalic size (EL/CS): 0.260 [0.239, 0.276]. Frontal carina distance vs. absolute cephalic size (FRS/CS): 0.409 [0.379, 0.430]. Longitudinal carinae on median region of frons: absent. Smooth median region on frons: absent. Antennomere count: 12. Scape length vs. absolute cephalic size (SL/CS): 0.659 [0.615, 0.694]. Facial area of the scape absolute setal angle: 15–30°. Median clypeal notch: absent. Ground sculpture of submedian area of clypeus: present. Median carina of clypeus: present. Lateral carinae of clypeus: present. Median anatomical line of propodeal spine angle value to Weber length in lateral view: 20–27°. Spine length vs. absolute cephalic size (SPST/CS): 0.310 [0.257, 0.356]. Minimum spine distance vs. absolute cephalic size (SPBA/CS): 0.369 [0.327, 0.405]. Apical spine distance vs. absolute cephalic size (SPTI/CS): 0.377 [0.337, 0.424]. Propodeal spine shape: triangular, blunt. Anterolateral pronotal corner: absent. Metanotal depression: present, inconspicuous. Dorsal region of mesosoma sculpture: rugulose with areolate ground sculpture. Lateral region of pronotum sculpture: areolate ground sculpture, superimposed by dispersed rugae. Mesopleuron sculpture: areolate ground sculpture, superimposed by dispersed rugae. Metapleuron sculpture: areolate ground sculpture, superimposed by dispersed rugae. Petiole width vs. absolute cephalic size (PEW/CS): 0.407 [0.363, 0.460]. Dorsal region of petiole sculpture: ground sculpture areolate, main sculpture dispersed rugose. Postpetiole width vs. absolute cephalic size (PPW/CS): 0.470 [0.429, 0.507]. Dorsal region of postpetiole sculpture: ground sculpture areolate, main sculpture dispersed rugose.

Etymology. This name fragilis (=fragile) refers to the small size of this species.

Diagnosis. Workers of N. fragilis differ from those of N. clypeatus by having no median clypeal notch (Fig. 4B) and from those of N. angulatus by the lack of an anterolateral pronotal corner (Fig. 4D). This species can be separated from N. devius, N. exiguus, N. gracilis, and N. hirtellus based on the MRA plot (Fig. 5A). A combination of two morphometric ratios (PPW/ML and CW/SPST) provides a classification success of 99.4% between N. fragilis and species of the N. devius complex. This species is the most similar to N. bidentatus, but can be separated using a PooC/SPTI ratio that yields 92.9% classification success (Fig. 5B). Due to the fact that the MRA plot offers only 97.3% of the correct classification, a reduced discriminant function using a combination of four characters (D4) that yields >99% classification success is also given for the most accurate determination in an area where the two species co-occur syntopically. Details are given in diagnosis under N. bidentatus.

Disrtibution. Nesomyrmex fragilis is distributed in tropical dry forests, disturbed forests, rainforests, and littoral rainforests in the Antsisarana region (Fig. 3). A single sample (CASENT0134410) is known to have been collected in the Mahajanga region far south of the main distributional area of this species. A wildcard test of this single sample confirmed its classification as N. fragilis (posterior p = 0.916).

Nesomyrmex gracilis Csősz & Fisher sp. n.

(Figs. 12A–12C, Table S1, Table 2.)

Figure 12 Nesomyrmex gracilis sp. n. holotype worker (CASENT0107191).

Head in full-face view (A), lateral view of the body (B), dorsal view of the body (C).

Type material investigated.

Holotype worker: MADAGASCAR: Prov. Antsiranana, Forêt Ambato, 26.6 km 33° Ambanja, 13°27.87′S, 48°33.10′E, 150 m, 8-11.xii.2004, collection code: BLF11548; CASENT0107191, Fisher et al. (CASENT0107191, CAS);

Paratypes: three workers, three gynes, and a male with the same locality data under CASENT codes: CASENT0763758, BLF11548, (1q, CAS); CASENT0107710, BLF11539, (1w, 1q, CAS); CASENT0107026, BLF11624, (1w, 1m, CAS); CASENT0107027, BLF11624, (1w, 1q, CAS);

Description of workers. Body color: yellow; brown. Body color pattern: concolorous. Absolute cephalic size (µm): 620 [508, 699], (n = 44). Cephalic length vs. maximum width of head capsule (CL/CWb): 1.199 [1.154, 1.246]. Postocular distance vs. cephalic length (PoOc/CL): 0.387 [0.372, 0.406]. Postocular sides of cranium contour frontal view orientation: parallel; converging anteriorly. Postocular sides of cranium contour frontal view shape: straight; feebly convex; convex. Vertex contour line in frontal view shape: straight; feebly convex. Vertex sculpture: main sculpture rugose, ground sculpture areolate. Gena contour line in frontal view shape: convex. Genae contour from anterior view orientation: converging; strongly converging. Gena sculpture: rugoso-reticulate with areolate ground sculpture. Concentric carinae laterally surrounding antennal foramen: absent. Eye length vs. absolute cephalic size (EL/CS): 0.252 [0.228, 0.274]. Frontal carina distance vs. absolute cephalic size (FRS/CS): 0.413 [0.390, 0.436]. Longitudinal carinae on median region of frons: absent. Smooth median region on frons: absent. Antennomere count: 12. Scape length vs. absolute cephalic size (SL/CS): 0.647 [0.622, 0.685]. Facial area of the scape absolute setal angle: 0–15°. Median clypeal notch: absent. Ground sculpture of submedian area of clypeus: present. Median carina of clypeus: present. Lateral carinae of clypeus: present. Median anatomical line of propodeal spine angle value to Weber length in lateral view: 40–45°. Spine length vs. absolute cephalic size (SPST/CS): 0.399 [0.301, 0.446]. Minimum spine distance vs. absolute cephalic size (SPBA/CS): 0.393 [0.350, 0.433]. Apical spine distance vs. absolute cephalic size (SPTI/CS): 0.489 [0.448, 0.536]. Propodeal spine shape: straight; slightly bent. Anterolateral pronotal corner: absent. Metanotal depression: present. Dorsal region of mesosoma sculpture: rugulose with areolate ground sculpture. Lateral region of pronotum sculpture: areolate ground sculpture, superimposed by dispersed rugae. Mesopleuron sculpture: areolate ground sculpture superimposed by dispersed rugulae. Metapleuron sculpture: areolate ground sculpture superimposed by dispersed rugulae. Petiole width vs. absolute cephalic size (PEW/CS): 0.454 [0.390, 0.487]. Dorsal region of petiole sculpture: ground sculpture areolate, main sculpture dispersed rugose. Postpetiole width vs. absolute cephalic size (PPW/CS): 0.500 [0.453, 0.539]. Dorsal region of postpetiole sculpture: ground sculpture areolate, main sculpture dispersed rugose.

Etymology. This name gracilis (=slender, slim) refers to the small, tiny appearance of this species.

Diagnosis. Workers of N. gracilis differ from those of N. clypeatus by having no median clypeal notch (Fig. 4B) and from N. angulatus by the lack of an anterolateral pronotal corner (Fig. 4D). This species can be separated from N. bidentatus and N. fragilis based on the apical spine distance ratio (SPTI/CS, see Table 2). This species occurs in the northern part of Madagascar syntopically wih N. exiguus from the N. devius complex. A simple ratio (PoOC/SPTI, see details in key) offers 94.5% success in distinguishing between this species and N. exiguus, and a combination of two ratios (PoOC/SPTI and CWb/ML) yields a safe determination (Fig. 5C). The other two species of this complex, N. devius and N. hirtellus, do not occur syntopically with this species, as both are distributed far south of the range of N. gracilis.

Distribution. This species is endemic to the Malagasy region, and its distribution is restricted to the northern, dry area of Madagascar (Fig. 3) in various habitats: rainforests, rainforest edges, littoral forests, and tropical dry forests.

Nesomyrmex hirtellus Csősz & Fisher sp. n.

(Figs. 13A–13C , Table S1, Table 2.)

Figure 13 Nesomyrmex hirtellus sp. n. holotype worker (CASENT0457484).

Head in full-face view (A), lateral view of the body (B), dorsal view of the body (C).

Type material investigated.

Holotype worker: MADAGASCAR: Prov. Toliara, Forêt de Beroboka, 5.9 km 131°SE Ankidranoka, 22.23306°S, 43.36633°E, 80 m, 12-16.iii.2002, collection code: BLF6155; CASENT0457484, Fisher et al. (CASENT0457484, CAS);

Paratypes: Twenty six workers and three gynes with the same locality data under CASENT codes:

CASENT0457483, BLF6155, (1w, CAS); CASENT0457482, BLF6155, (1w, CAS); CASENT0457476, BLF6155, (1w, CAS); CASENT0457481, BLF6155, (1q, CAS); CASENT0457480, BLF6155, (1q, CAS); CASENT0457478, BLF6155, (1w, CAS); CASENT0457479, BLF6155, (1w, CAS); CASENT0457477, BLF6155, (1w, CAS); CASENT0439545, BLF6118, (1w, CAS); CASENT0439552, BLF6118, (1w, CAS); CASENT0439546, BLF6118, (1w, CAS); CASENT0439555, BLF6118, (1w, CAS); CASENT0439551, BLF6118, (1w, CAS); CASENT0439547, BLF6118, (1w, CAS); CASENT0439553, BLF6118, (1w, CAS); CASENT0439548, BLF6118, (1w, CAS); CASENT0439550, BLF6118, (1w, CAS); CASENT0439549, BLF6118, (1w, CAS); CASENT0439554, BLF6118, (1w, CAS); CASENT0457598, BLF6119, (3w, CAS); CASENT0457596, BLF6119, (1q, CAS); CASENT0457599, BLF6119, (3w, CAS); CASENT0457597, BLF6119, (3w, CAS);

Description of workers. Body color: yellow; brown. Body color pattern: concolorous. Absolute cephalic size (µm): 592 [525, 641], (n = 75). Cephalic length vs. maximum width of head capsule (CL/CWb): 1.187 [1.142, 1.242]. Postocular distance vs. cephalic length (PoOc/CL): 0.387 [0.366, 0.404]. Postocular sides of cranium contour frontal view orientation: parallel; converging anteriorly. Postocular sides of cranium contour frontal view shape: straight; feebly convex; convex. Vertex contour line in frontal view shape: straight; feebly convex. Vertex sculpture: main sculpture rugose, ground sculpture areolate. Gena contour line in frontal view shape: convex. Genae contour from anterior view orientation: converging; strongly converging. Gena sculpture: rugoso-reticulate with areolate ground sculpture. Concentric carinae laterally surrounding antennal foramen: absent. Eye length vs. absolute cephalic size (EL/CS): 0.272 [0.249, 0.289]. Frontal carina distance vs. absolute cephalic size (FRS/CS): 0.412 [0.385, 0.427]. Longitudinal carinae on median region of frons: absent. Smooth median region on frons: absent. Antennomere count: 12. Scape length vs. absolute cephalic size (SL/CS): 0.667 [0.611, 0.705]. Facial area of the scape absolute setal angle: ca. 15°. Median clypeal notch: absent. Ground sculpture of submedian area of clypeus: present. Median carina of clypeus: present. Lateral carinae of clypeus: present. Median anatomical line of propodeal spine angle value to Weber length in lateral view: 37–42°. Spine length vs. absolute cephalic size (SPST/CS): 0.375 [0.332, 0.416]. Minimum spine distance vs. absolute cephalic size (SPBA/CS): 0.389 [0.345, 0.427]. Apical spine distance vs. absolute cephalic size (SPTI/CS): 0.460 [0.418, 0.504]. Propodeal spine shape: straight; slightly bent. Anterolateral pronotal corner: absent. Metanotal depression: present. Dorsal region of mesosoma sculpture: rugulose with areolate ground sculpture. Lateral region of pronotum sculpture: areolate ground sculpture, superimposed by dispersed rugae. Mesopleuron sculpture: areolate ground sculpture superimposed by dispersed rugulae. Metapleuron sculpture: areolate ground sculpture superimposed by dispersed rugulae. Petiole width vs. absolute cephalic size (PEW/CS): 0.460 [0.409, 0.522]. Dorsal region of petiole sculpture: ground sculpture areolate, main sculpture dispersed rugose. Postpetiole width vs. absolute cephalic size (PPW/CS): 0.525 [0.475, 0.585]. Dorsal region of postpetiole sculpture: ground sculpture areolate, main sculpture dispersed rugose.

Etymology. The name hirtellus: hirtus (=hairy) + -ellus (diminutive) refers to the workers having short hairs.

Diagnosis. Workers of N. hirtellus differ from those of N. clypeatus by having no median clypeal notch (Fig. 4B) and from N. angulatus by the lack of an anterolateral pronotal corner (Fig. 4D). This species can be separated from N. bidentatus and N. fragilis based on the apical spine distance ratio (SPTI/CS, see Table 2) with a single misclassified N. fragilis individual. This species occurs in the middle-western and southern part of Madagascar syntopically with N. devius from the N. devius complex. A single ratio (PoOC/ SPST, see details in key) offers 92.2% success in distinguishing between this species and N. devius and a combination of two ratios (PoOC/SPST and MW/PPH) yields a safe determination (Fig. 5D). The other two species of this complex, N. exiguus and N. gracilis, do not occur syntopically with this species, as both are distributed far to the north of the range of N. hirtellus.

Distribution. This species is endemic to the Malagasy region, and its distribution is restricted to the southwestern, sub-arid area of Madagascar (Fig. 3); it can be found in tropical dry forest and spiny forest.

Conclusion

Combined application of exploratory techniques NC-PART clustering on continuous morphological data revealed that the N. angulatus species-group comprises eight well-outlined clusters in the Malagasy zoogeographical region, all representing species; of these, seven taxa are new to science. Delimitations of clusters recognized by the currently introduced combination of morphometric procedure NC-PART clustering were tested via confirmatory Linear Discriminant Analysis (LDA) and Multivariate Ratio Extractor, MRA (Baur & Leuenberger, 2011).

The fusion of NC clustering and partitioning method PART combines the advantages of the two methods. The dimensionality reduction feature and hierarchical visual display of NC-clustering (Seifert, Ritz & Csősz, 2014) are reinforced by the partitioning method PART, which makes assignments to objectively-defined clusters based on statistical thresholds (Nilsen et al., 2013; Tibshirani, Walther & Hastie, 2001).

The most important benefit of this procedure is that cluster assignments are no longer user-defined, but rather done by the algorithm based on statistical criteria; this is in contrast to hierarchical clustering, where one must decide the boundaries of meaningful clusters. The new, combined method not only offers greater automatization and produces faster inferences, but also takes another step toward increased objectivity in the decision making process of morphometry-based alpha taxonomy.

Quantitative morphometric approaches are often misinterpreted as causing oversplitting as a result of excessive discriminatory power, a characteristic wrongly attributed to these algorithms. In exceptional cases, multivariate statistics on morphometric data may lead to oversplitting, particularly if the cluster boundaries are not properly defined. However, the combined NC-PART clustering is method prevents unjustified oversplitting i.e., treating precarious sub clusters as meaningful fragments via application of the gap statistic criterion (Csősz & Fisher, 2015; Tibshirani, Walther & Hastie, 2001) implemented in method PART.

The combined procedure presented here is a valuable asset to morphometry based alpha taxonomy as it provides greater resolving power, increased objectivity, and largely automated decision making. However, they do not believe that the validity of the patterns obtained by the new procedure would be exclusive or superior to alternative solutions when additional biological information is available, such as molecular data, discrete morphological data, distribution, natural history. To the contrary, the authors hold that diverse evidence from different approaches is necessary to achieve the highest quality measures of biodiversity. The current method contributes to the analysis of complex morphometric data in a manner allows for increased objectivity and independent hypothesis formation in the taxonomic workflow.

Supplemental Information

Table S1 List of morphometrically investigated samples

Unique CASENT number for pinned samples, locality, geographic coordinates (Latitude, Longitude) in decimal format, altitude (Elevation) in meters a.s.l., collector’s name, date and number of specimens investigated bearing the given CASENT number are provided. HT, Holotype; PT, paratype(s).

Click here for additional data file.

Table S2 URI table for morphometric characters

Hymenoptera-specific terminology of morphological statements used in descriptions, identification key, and diagnoses are mapped to classes in phenotype-relevant ontologies.

Click here for additional data file.

Table S3 Morphometric data for samples of the angulatus species group

Morphometric data of 21 continuous morphometric traits of 378 individuals, plus two others applied to a small fragment of samples, is given in µm. CASENT code (casent), final species hypothesis (species), geographic coordinates (long, lat) and the name format as samples appear on the dendrogram (dendro-name) are also provided in the table. HT, Holotype; PT, paratype(s).

Click here for additional data file.

Supplemental Information 1 R script of NC-PART clustering including mark.dendrogram

R script of NC clustering and method PART implementing cluster methods “hclust” and “kmeans”. Mark dendrogram function mapping the results of partitioning algorithm PART on the dendrogram is also added.

Click here for additional data file.

Additional Information and Declarations

Competing Interests

Author Contributions

Field Study Permissions

Data Availability

New Species Registration

The authors declare there are no competing interests.

Sándor Csősz conceived and designed the experiments, performed the experiments, analyzed the data, contributed reagents/materials/analysis tools, wrote the paper, prepared figures and/or tables, reviewed drafts of the paper.

Brian L. Fisher contributed reagents/materials/analysis tools, wrote the paper, reviewed drafts of the paper.

The following information was supplied relating to field study approvals (i.e., approving body and any reference numbers):

Ant samples used in this study comply with the regulations for export and exchange of research samples outlined in the Convention of Biology Diversity and the Convention on International Trade in Endangered Species of Wild Fauna and Flora. For field work conducted in Madagascar, permits to research, collect and export ants were obtained from the Ministry of Environment and Forest as part of an ongoing collaboration between the California Academy of Sciences and the Ministry of Environment and Forest, Madagascar National Parks and Parc Botanique et Zoologique de Tsimbazaza. Authorization for export was provided by the Director of Natural Resources.

Approval Numbers:

No. 0142N/EA03/MG02,

No. 340N-EV10/MG04,

No. 69 du 07/04/06,

No. 065N-EA05/MG11,

No. 047N-EA05/MG11,

No. 083N-A03/MG05,

No. 206 MINENVEF/SG/DGEF/DPB/SCBLF,

No. 0324N/EA12/MG03,

No. 100 l/fEF/SG/DGEF/DADF/SCBF,

No. 0379N/EA11/MG02,

No. 200N/EA05/MG02.

The following information was supplied regarding data availability:

The raw data is supplied as a Supplemental Information.

The following information was supplied regarding the registration of a newly described species:

Nesomyrmex bidentatus sp. nov. lsid:zoobank.org:act:EE82C681-40E5-471F-8324-8C8B4552F7CF;

Nesomyrmex clypeatus sp. nov. lsid:zoobank.org:act:3EA024EF-600A-4809-81B0-C370DE53841F;

Nesomyrmex devius sp. nov. lsid:zoobank.org:act:B624AD66-3385-4544-8381-A6D25C5DD064;

Nesomyrmex exiguus sp. nov. lsid:zoobank.org:act:D125DEA9-F3A7-4B1A-A51B-EA7284DC81C6;

Nesomyrmex fragilis sp. nov. lsid:zoobank.org:act:E85A138F-4ABB-4C6A-989D-8DF0E6A83EFE;

Nesomyrmex gracilis sp. nov. lsid:zoobank.org:act:9AA4EAD6-3483-41E7-8AF8-229B84716BBA;

Nesomyrmex hirtellus sp. nov. lsid:zoobank.org:act:8521AD95-9948-42B6-905F-7A1D43DDF4F8.

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
