# Peer review of "Taxonomic revision of the Malagasy members of the Nesomyrmex angulatus species group using the automated morphological species delineation protocol NC-PART-clustering"

_PeerJ, doi:10.7717/peerj.1796_

## Round 0.1 · original submission · Minor Revisions

· Academic Editor

Minor Revisions

All of the reviewers find the study valuable and well executed, although they suggest that the writing, particularly the introduction, can be improved. I believe that all of the concerns can be easily addressed in a minor revision.

As pointed out by the Reviewer 2, the authors should provide all of the raw data and analysis scripts necessary to reproduce this study. This is also required under PeerJ policy: https://peerj.com/about/policies-and-procedures/

Reviewer 1 ·

Basic reporting

The article investigates species limits in a compelx of ants from Madagascar. The authors do a nice job with the taxonomic revision of the group, and in describing the new species to science. The work is thorough, and the descriptions and images of the new species are readily available.

The manuscript, however, could use more detail. For example, the Introduction is short and is similar in structure to an Abstract. I find it lacking in presenting information as to what has driven this study. Why study this group of ants? Is this a difficult group to determine species limits based on traditional analysis of morphological data? Was sampling too sparse before this study? What about general information about ants in Madagascar? Overall, there is little background information on the study system and geographic region. The manuscript would be improved if the Introduction was expanded, and included more background information on Madagascar and ants in the region.

Experimental design

I found the experimental design to be sufficient for the study. I did find, however, that a few of the methods lacked sufficient detail.

-(lines 105-106). What function was used in phytools to make the distribution maps?

-(lines 220-221) What functions were used in the two R packages? Did you have to specify certain criteria to run those analyses?

-(lines 223-224) What criteria were used in the function ‘part’? Did you specify a maximum number of clusters allowed? A minimum number of objects required for a cluster?

In general, I found it challenging to follow the specifics of the analyses, and thus, would have a tough time trying to recreate the approach for species delimitation. I would recommend being more specific with what you did, what criteria were set in the R functions and other methods, so that other researchers can follow your approach when applying these techniques to their taxonomic groups.

Validity of the findings

The findings are supported by the data, and the species descriptions are well done.

Additional comments

The results are presented clearly, but there is no discussion as to what these results mean. I would recommend the authors to expand upon their findings, talk about how their approach to species delimitation is an improvement to previously used approaches, and why this approach allowed them to disentangle species limits in a group where this was previously unable to be done. I would also recommend for the authors to add a Conclusion section, and bring home the main points of the paper. Two additional comments:

I found the paragraph from lines 251-256 to be vague and lacking in support. Based on the maps in Figure 3, many of the species are overlapping in distribution. Isn’t that in conflict with the sentence on line 251? Also, where are the data that show the species occupy different niches? How was niche space quantified and compared? Please provide evidence to justify this statement about niches.

Figure 3 – As drawn, N. angulatus has samples in the ocean. You may want to state in the Figure legend that those samples are from surrounding islands.

·

Basic reporting

NC-clustering and partitioning algorithm based on recursive thresholding, the two methods on which the paper is based, are explained but the novelty and significance of using them in combination is not apparent. How does the combination of the two methods present an advance in morphometry-based species delimitation? Why is this approach superior to other similar methods and has it been used in taxonomy before?

The availability of raw data is insufficient. The conclusions of the paper are based on extensive analysis of morphometric data using multiple R packages. The authors reference a paper in review that is supposed to provide some of the R code used to generate data in this contribution, but since the scripts are not available with this submission, I was not able to access them. The scripts must be available to the reviewers and ideally one should be able to replicate all the results by using supplementary code and raw morphometric data provided with the manuscript.

The names of terminals and the vertical right-hand side dendrogram title are difficult to read in Figure 2. In the final submission at the very least the resolution should be higher. The same goes for Figure 3; it seems that tip labels could be dropped from the dendrogram since species names are indicated by colors. By the way, the colors used in figures would also difficult to read for some color-blind people. It's not difficult to find color-blind friendly palettes and several are available for R.

Finally, I don't see an explanation for the treatment of the outliers that are obvious in Figure 2. How they were incorporated into the final species hypothesis?

Experimental design

No comments.

Validity of the findings

No comments.

Additional comments

This manuscript is an example of modern high-quality research in ant taxonomy. The authors present a mature attempt at automated clustering of samples using a statistically-explicit approach. The manuscript is well-written and data is well presented (except the lack of computational scripts and some issues with figures, as mentioned above). In addition to the apparently clustering approach novel in this context, the authors constructed taxonomic descriptions using an ontology. Overall I think this contribution definitely deserves publication.

One final suggestion I have relates to the underlying philosophy of the approach. Morphometric-based approaches such as this, even using very sophisticated clustering approaches, are in line with the so-called typological species concepts. The discovery of gaps in the phenotype space combined with the assertion that this approach represents "objective" approach to taxonomy deemphasizes the importance of biological processes underlying the variation. The authors mention that recognized species also differ in geographical range and ecological preferences. Does this mean that they claim the taxonomic hypotheses created using clustering reflect biological species? If that is the case, there should be a set of other criteria more directly linked to speciation process (sympatric occurrence, reproductive isolation) against which clustering methods could be evaluated. Otherwise the authors should explicitly identify their approach as following the typological species concept.

·

Basic reporting

- The English should be reviewed by a native English speaker. A few sentences are unclear due to incorrect word choice and articles are often inappropriately omitted, hindering clarity.
- The introduction is written like an abstract. It includes background information as well as the results and conclusions of the paper. I would suggest moving the conclusions to the results and discussion section.
- The introduction does not introduce the taxonomic group of focus adequately. It provides adequate background information about the methods, but lacks information on the taxonomic history of Nesomymrex in Madagascar. Such taxonomic information is essential to any revisionary work. At the very least another paper should be cited clearly.

Experimental design

- As mentioned in the "basic reporting" section, the taxonomic problem (= research questions) is not adequately reviewed in the introduction. The authors should provide some background information about the systematics of Nesomyrmex in Madagascar and/or globally.
- The introduction implies that the authors are testing the NC-Part approach to species delimitation. I would suggest the authors include more information in the discussion about the value of the approach as compared to more traditional, less quantitative taxonomy.
- The authors should mention in the methods if species were initially delimitated by traditional morphological examination. Similarly, did the quantitative approach reveal any cryptic species that were not apparent by eye? Is the value of the method simply that it reduces error rates in identification or does it improve initial species delimitation? The latter might be good fodder for the discussion.

Validity of the findings

No Comments

Additional comments

Overall this is an excellent paper. The quantitative methods are progressive, helping to push systematics into the future, and the basic taxonomic work (key, descriptions, figures, etc.) is thorough and well done. In addition I commend the authors for making all of the specimen data and holotype images available online. This really benefits taxonomy by having the primary data available for anyone to review.

My main negative comment is that the introduction reads too much like an abstract and focuses too heavily on the quantitative methods. The introduction should only introduce the method, leaving a discussion of the results for the discussion section. The intro should also mention the current state of Nesomyrmex taxonomy and something about the biology of the group. If this information is detailed elsewhere, then please make the references more obvious.

The species descriptions would benefit from having a review of any known biological information. This information can be gleaned from specimen data on antweb, but having a review of the information, written by the author, helps one to quickly see the dominant characteristics of the species.

The det labels imaged on AntWeb for type specimens list the year as "2015". To avoid confusion, these should be updated to 2016.

Minor Comments:
L55: Change "Csősz & Fisher 2016" to "in press" or "in review" until it is published.
L58: Change "proved" to "proven"
L80: Change "Convention of Biology Diversity" to "Convention on Biological Diversity"
L89-92: This paragraph is unclear and perhaps in a poor position. First, I believe you need to change "2 continuous" to "24 continuous". Second, it would make more sense to move this paragraph down to the discussion of measured characters.
L113-114: The sentence "Worker-based revision..." is unclear and should be revised.
L389: Delete extra "G" in Madagascar.
L288-290: The sentence "Classification..." is unclear and should be revised. Did you mean to write "parentheses" in place of "brackets"?

---

## Round 0.2 · Minor Revisions

· Academic Editor

Minor Revisions

Overall the authors did a good job on the revision, but the requested files, which are supposed to be text files, are binary files instead - the raw data and (S3) and the script (S4). Please check your submission.

---

## Round 0.3 · accepted · Accept

· Academic Editor

Accept

I feel that the authors have done a satisfactory job replying to the reviewers, and the supplementary data look OK now. I recommend this manuscript for publication.